# Psychological State after an Acute Coronary Syndrome: Impact of Physical Limitations

**DOI:** 10.3390/ijerph18126473

**Published:** 2021-06-15

**Authors:** Miguel-Ángel Serrano-Rosa, Eva León-Zarceño, Cristina Giglio, Salvador Boix-Vilella, Antonio Moreno-Tenas, Lidia Pamies-Aubalat, Vicente Arrarte

**Affiliations:** 1Department of Psychobiology, University of Valencia, 46010 Valencia, Spain; m.angel.serrano@uv.es (M.-Á.S.-R.); crisgiglio95@gmail.com (C.G.); 2Departamento de Ciencias del Comportamiento y Salud, Universidad Miguel Hernández de Elche, 03202 Elche, Spain; antonio.morenot@umh.es (A.M.-T.); lpamies@umh.es (L.P.-A.); 3Facultad de Humanidades y Ciencias Sociales, Universidad Isabel I. Burgos, 09003 Burgos, Spain; salvador.boix@ui1.es; 4Cardiology Unit, Hospital of Alicante, 03010 Alicante, Spain; viarrarte@hotmail.com

**Keywords:** acute coronary syndrome, physical limitations, psychological factors, patient perception, quality of life

## Abstract

The aim of this study was to investigate how physical limitations after ACS influence patients’ quality of life and health perception. This was a longitudinal clinical study. We recruited 146 patients diagnosed with ACS. The patients performed a stress test (Bruce’s protocol) for the evaluation of physical limitations and were classified according to the test result: without physical limitations (more than 10 METS), with some physical limitations (7 to 9 METS), and with high physical limitations (less than 6 METS). Significant differences were found between the three groups immediately after the diagnosis of ACS and after a period of three months, regarding health perception, anxiety, depression, sexual relationships, distress, and adjustment to disease. These differences resulted larger between the group with less limitations and the group with higher limitations. After 3 months, however, there was an overall improvement in all variables. In conclusion, physical limitations after ACS seem to influence perceived quality of life determined by measuring general health, vitality, total adaptation, emotional role, social adaptation, depression, and anxiety. Therefore, the highest the physical limitations, the poorer the psychological conditions and vice versa, even 3 months after ACS diagnosis.

## 1. Introduction

The term acute coronary syndrome (ACS) covers a wide range of clinical situations, from unstable angina to myocardial infarction with ST segment elevation (STEMI). These are, with rare exceptions, a consequence of the formation of acute clots linked to an interrupted coronary atherosclerotic plaque [1]. ACS is included in the broader category of cardiovascular disease (CVD). CVD causes about one-third of all deaths worldwide, of which an estimated 7.5 million deaths are due to ischemic heart disease (IHD). ACS and sudden deaths cause most IHD-related deaths, amounting to 1.8 million deaths per year. The incidence of IHD in general and linked to ACS increases with age, although, on average, it occurs 7–10 years earlier in men than in women [2]. ACS occurs much more often in men than in women under 60 years of age, but most patients over 75 years of age are women, probably because women outlive men. The risk of acute coronary events in life is linked to exposure to traditional cardiovascular risk factors. Although the number of deaths from ACS has reduced in recent years as a result of prevention and treatment, this trend could be reversed due to the ageing population and the increase in some risk factors [3].

Although many factors predispose to the risk of acute coronary syndrome, there are many psychological and non-psychological symptoms that arise after the disease [4] that should be taken into account to care for these patients. Psychosocial features influenced by ACS include depression, anxiety, and self-perceived health (Huffman, 2010). These are some of the most studied psychological factors considered to be a consequence of ACS. However, are these factors a direct consequence of acute coronary syndrome, or are there other factors that should be taken into account to explain the effects that ACS has on psychological variables?

After ACS, some physical limitations appear that depend on the physical state of the patient. Thus, before discharge from the hospital, patients are informed that in their daily life they should limit their efforts depending on their physical state, assessed by an ergometric approach, such as the Bruce protocol. Physical effort is measured using the metabolic equivalent of a task (METS) [5]. Thus, based on the test result, patients can be classified according to their degree of physical function. In particular, the World Health Organization (WHO) establishes three levels of functional capacity according to the cardiac stress test after ACS. Patients exceeding 10 METS are considered without physical limitations, patients under 6 METS have severe limitations (they can only sustain light exertion or sedentary work), and those between 7 and 9 METS are at an intermediate level with some physical limitations (cannot sustain heavy exertion, but only light exertion). Cardiologists then instruct patients on the type of efforts patients can make in their daily life, which may result in some physical limitations and, therefore, difficulties in adapting to their family and social environment. However, despite these instructions, to our knowledge, there are few studies focusing on how these limitations influence psychosocial variables. Thus, it has recently been suggested that better physical capacity after ACS is associated with higher quality of life scores after cardiac rehabilitation, although these associations are weak [6]. This study aims to describe how limitations, classified according to the results of the Bruce test, influence quality of life, determined by measuring social functioning, psychosocial adaptation to disease, anxiety, depression, and personal growth.

### Background

It is well known that functional aspects of behavioral and physical health are of fundamental importance for patients who have suffered from ACS. Patients often face limitations imposed by the disease, but there are important psychological consequences that affect their quality of life [7]. These factors need to be considered because they influence adherence to treatments and patients’ quality of life [6,8,9,10].

As previously mentioned, psychosocial factors affected by ACS include depression, anxiety, and self-perceived health situation [11,12,13]. Specifically, depression and anxiety are influenced negatively by the cardiac outcome of patients after ACS both in the hospital and, above all, in the long term. In fact, depression is three times more common in patients who have suffered from ACS than in the general population. Depressive symptoms are associated with recurrent cardiac events and mortality, as well as poor quality of life [12]. Despite their strong impact, these symptoms are often not recognized nor treated and may persist for weeks, months, or years [11]. In addition, posttraumatic stress has also been described in these patients, because of the immediate and unexpected nature with which the disease manifests itself [14]. The latter increases the risk of recurrence and mortality [15].

Moreover, from a psychosocial point of view, cardiac events increase the risk of poorer professional conditions including reduced responsibility, part-time employment, lower salary, and discharge from jobs with a mean productivity loss [16]. No strong relationships have been found between patients’ functional status after ACS and the severity of ACS [17], and some psychosocial variables (stressful events or social support) have been proposed as possible predictors [18].

Furthermore, ACS has been associated with a lower quality of life after the event [19]. In particular, it has been shown that physical functions (physical function, physical role, body pain, and general health) are reduced after the cardiac event [20], and quality of life after one year may depend on the type of cardiac intervention [21]. Thus, physical activity in daily life is altered in the presence of diseases or injuries, as cardiac events often lead to alterations in the sphere of physical functionality and mental health [22].

Finally, after an ACS, many life changes occur that can make patients reconsider their life. In this sense, individual differences may influence the evolution of life (quality and health) after ACS. However, to our knowledge, there are no studies that analyze or report whether in some patients there may be personal growth or a change in life perspective after ACS.

Considering the physical limitations often faced by patients with ACS, the aim of this study was to investigate the impact of physical functionality limitation on psychological and mental health in patients suffering from ACS. We hypothesized that the more extensive the physical limitations, the greater the psychological and social consequences, especially regarding anxiety and depression, quality of life (physical health and mental health) adjustment to disease (attitude to health, distress, family, social, domestic, sexual, professional relations), and posttraumatic growth (change appreciation, relationship with others, personal strength, spiritual change). Our second hypothesis was that patients with less physical limitations would have a better psychological evolution after three months.

## 2. Materials and Methods

### 2.1. Design

The design of the study was cross-sectional, with a two-point data collection with respect to functional and psychological aspects of ACS patients, who were classified according to their degree of functional limitation through stress testing. Patients were evaluated in two phases: phase 1 (after admission to hospital with a diagnosis of ACS) and phase 2 (three months after discharge from hospital).

### 2.2. Sample/Participants

In this study, 146 patients (mean age = 57.86, SD = 9.45) participated. All were recruited by the cardiology unit of the hospital of Alicante (Spain) after being admitted in the hospital and diagnosed with ACS. They signed an informed consent approved by the ethical commission of the General Hospital of Alicante. Once patients were operated and stabilized, they were invited to participate in the study and, once accepted, they signed an informed consent to participate in the study.

### 2.3. Data Collection

The medical history of each patient was provided by the medical record that reported the diagnosis of cardiac comorbidity such as hypertension, hyperlipidemia, obesity, and previous cardiac events. Furthermore, body mass index (BMI), blood pressure, heart rate, cholesterol, triglyceride levels, glomerular filtration, and glycate levels were considered. Finally, the days of stay in the hospital were counted. Once these parameters were obtained, patients performed an effort stress test (Bruce protocol) in order to evaluate their physical functioning to determine the physical limitations that these patients would have in their daily life.

For the classification of patients depending on physical limitations, an effort stress test (Bruce protocol) was used. The Bruce protocol is one of the most frequently performed non-invasive tests to study patients with suspected coronary heart disease and to determine prognosis and functional ability in patients with proven coronary heart disease [23]. This protocol involves recording an electrocardiogram and measuring blood pressure during physical exertion. In the treadmill stress test, the patient walks on a treadmill, with speed and slope increased every 3 min. The exercise test is suspended when the objectives of the test are achieved (the theoretical maximum heart rate is reached) or when there are clinical and electrocardiographic signs that make it advisable to stop the test for the safety of the patient. The latter possibility includes physical fatigue, progressive angina, dizziness or instability, a progressive drop in blood pressure, an excessive increase in blood pressure (>250/130 mmHg), major changes in heart rhythm, ST depression >4 mm, or ST increase >2 mm in the absence of a previous heart attack. This level of effort is measured using the metabolic equivalent (METS) [5]. Based on the test result, patients can be classified into three groups according to their degree of physical activity: patients that with more than 10 METS (group A) are considered without physical limitations, those scoring between 7 and 9 METS (group B) are at an intermediate level with some physical limitations (cannot perform hard efforts but only light efforts), and those scoring under 6 METS (group C) have serious limitations (they can perform only light efforts).

Following the preliminary tests, the patients filled in questionnaires for the evaluation of the psychological variables. Three months after discharge from the hospital, the patients filled in the questionnaires again to verify their psychological state. Therefore, all variables were measured two times (after ACS and after three months).

As regards the assessment of the psychological aspects, the following questionnaires were used:

SF-12 (Short-Form Health Survey) is a questionnaire that aims to investigate the perception of the psychophysical conditions of individuals. The SF-12 measures eight factors: physical functioning, role limitations due to physical health problems, bodily pain, general health, vitality (energy/fatigue), social functioning, role of limitations due to emotional problems, and mental health (psychological distress and psychological well-being). From these measures, two indexes are used: the Physical Component Summary (PCS) index that covers the physical status, and the Mental Component Summary (MCS) index that measures the mental status [24].

HADS (Hospital Anxiety and Depression Scale) is a self-report questionnaire commonly used to determine the levels of anxiety and depression in physically ill patients. HADS consists of 14 items, 7 of which refer to anxiety, and 7 to depression [25].

PAIS (Psychosocial Adjustment of Illness Scale) is a multi-dimensional, semi-structured clinical interview designed to assess the psychological and social adjustment of medical patients, or of members of their immediate families, to the patients’ illness. The PAIS comprises 46 items and provides information along 7 domains of adjustment (health care orientation, vocational environment, domestic environment, sexual relations, extended family, social environment, and psychological distress) with an acceptable degree of reliability and validity [26].

The Posttraumatic Growth Inventory (PTGI) [27] is a 21-item self-assessment measure that determines the degree of positive change in the fight against major life crises. Items refer to positive changes in post-traumatic situations (i.e., “A sense of self-confidence” and “the willingness to express my emotions”). All items have a Likert scale response format ranging from 0 to 5 points. The analysis of the factors of the original instrument evaluates four areas of post-traumatic growth: (1) better appreciation of life and changing personal goals and priorities; (2) improvement in interpersonal relationships and a greater sense of intimacy with close ones; (3) a greater sense of strength and personal potentialities; (4) changes in spirituality [27]. The inventory has an acceptable construction validity, internal consistency (0.90), and test–test reliability over a 2-month interval (0.71).

### 2.4. Ethical Considerations

The study was carried out in accordance with Law 14/2007 of 3 July (Helsinki Declaration, 2008). All patients included in the practice read the patient information sheet and signed the informed consent. All patient data were anonymized by assigning a double code for both the sample and the data file, and only properly authorized personnel had access to personally identifiable information. The protocol of this study was approved by the ethical commission of the General Hospital of Alicante (Spain) (Ref. CEIC PI2013/31).

### 2.5. Data Analysis

Normality criteria were not met for the variables; therefore, non-parametric tests were used. Kruskal–Wallis test was used to perform comparisons within groups, and the Mann–Whitney test to compare two groups. When comparing variables within a group, the Wilcoxon test was performed. All Means (M) and Standard Deviations (SD) and statistical tests are summarized in tables. Finally, all analyses were performed with SPSS 24.0 (IBM Corporation, Armonk, NY, USA), with a significant level of 0.05.

### 2.6. Validity and Reliability of Measures

All measures used in this study are clinically validated measures [24,25,26,27]. Specifically, the reliability indexes for the measure of this study were the following: for SF-12 after ACS, the reliability indexes were 0.79 for the physical dimension and 0.88 for the mental dimension. After three months, Cronbach alpha was 0.76 (physical dimension) and 0.85 (mental dimension). For PAIS inventory, the reliability index was 0.91 for both measures (after ACS and after three months). For HAD, the reliability index obtained for anxiety was 0.86, while that for depression was of 0.83. After three months, Cronbach alpha was 0.85 and 0.82, for anxiety and depression, respectively. PTGI obtained a Cronbach alpha of 0.96 for both measures.

## 3. Results

### 3.1. Preliminary Results

There were no differences within groups for previous stroke, psychiatric disease, obesity, heart events, dyslipidemia, or previous neoplasm. Only differences in hypertension were present, with group A presenting less cases than the other groups (*p* = 0.014). Moreover, there were no significant differences in duration of the hospital stay, BMI, blood pressure, heart rate, and cholesterol level. However, differences in age (*p* < 0.001), triglyceride levels, and cholesterol–LDL levels were found. Specifically, participants in group A were younger than patients in groups B and C (*p* < 0.001). In the case of triglycerides and cholesterol LDL levels, differences were found (*p* = 0.023 and *p* = 0.009, respectively) only between groups A and C (see Table 1).

### 3.2. Differences in Social functioning, Adjustment to Disease, Anxiety, Depression, and Posttraumatic Growing between Groups

Table 2 contains the means and SD of all variables after ACS and after three months from ACS, including the significant differences. Furthermore, to facilitate reading of the results, statistics were included in the table. In the text, only differences observed in exclusively one of the two measures are specified. If no specification is made, it means that both measures were significantly different.

#### 3.2.1. Social Functioning (SF-12)

Differences were observed within groups for the following variables: pre- and post-ACS general health, physical function, physical role, emotional role, body pain, vitality, social function, and mental health. Therefore, for the variables of physical health and mental health, there were significant and consistent differences within groups immediately after ACS and three months after it (*p* < 0.001).

Post-hoc analyses indicated differences between groups A and B in body pain (U = 981.500, *p* = 0.044 and U = 992.000, *p* = 0.048, respectively).

Moreover, when groups A and C were compared, differences appeared for all variables of SF-12 (*p* < 0.01).

Finally, differences between groups B and C were observed for physical function (U = 144.000, *p* = 0.000 and U = 197.500, *p* = 0.020), physical role (U = 219.000, *p* = 0.048 and U = 160.500, *p* = 0.001), emotional role (U = 219.00, *p* = 0.044 and U = 201.500, *p* = 0.018), body pain (after three months; U = 214.000, *p* = 0.044), vitality (U = 189.000, *p* = 0.014 and U = 189.500, *p* = 0.014), social function (U = 181.000, *p* = 0.007 and U = 218.500, *p* = 0.05), mental health after three months (U = 203.000, *p* = 0.032), and general PCS (U = 173, *p* = 0.007 and U = 162.000, *p* = 0.003) and MCS (U = 180.500, *p* = 0.011 and U = 164.500, *p* = 0.004).

#### 3.2.2. Adjustment to Disease (PAIS)

In the case of adjustment to disease, differences were observed within groups in sexual relations, distress, and adjustment to disease (at both times). Post hoc analyses showed no differences between groups A and B. However, groups A and C differed in regard to sexual relations, distress, and total adjustment to disease. In addition, group B differed from group C only in sexual relations after ACS, but not for the other variables. Therefore, the main differences were found between groups A and C.

#### 3.2.3. Anxiety and Depression (HAD)

There were differences in anxiety and depression within groups at both times of measurement. Specifically, groups A and B differed in depression after ACS, and, in the case of anxiety, differences were marginal (*p* < 0.07) after ACS and after 3 months from ACS. Scores of groups A and C for anxiety and depression were different at both times. However, there were no significant differences between groups B and C for any variables at both times.

#### 3.2.4. Posttraumatic Growth (PTGI)

For posttraumatic growth after ACS, there were no differences between groups at both times.

### 3.3. Changes Immediately after ACS and Three Months after ACS by Group

Table 3 presents the statistics about this section.

#### 3.3.1. Group A (<10 METS)

After three months, participants in group A significantly improved their scores for almost all variables of SF-12: general health, physical functioning, vitality, mental health, and for the summed variables physical health and mental health. Furthermore, these patients improved their scores for some variables of PAIS: health attitude, domestic context, sexual relations, distress, and adjustment to disease. In addition, scores for anxiety and depression (from HAD) significantly improved after three months. Finally, for posttraumatic growth, this group significantly increased their scores for life appreciation, social relations, and personal strength.

#### 3.3.2. Group B (7–9 METS)

In the case of group B, participants did not significantly improve their scores of SF-12. However, based on data obtained from the PAIS questionnaire, there were significantly higher scores for health attitude, distress, and total adjustment after three months. Anxiety was significantly reduced after three months, while depression scores remained steady. Finally, this group increased the score for life appreciation in the posttraumatic growing inventory, while the scores in the rest of the scales did not change significantly.

#### 3.3.3. Group C (>7 METS)

In the case of patients in group C, physical function and general health scores (from SF-12) increased significantly after three months. Moreover, health attitude, distress, and adjustment to disease scores were higher after three months. In addition, anxiety and depression scores increased significantly in this group. However, there were not significant changes in ICP scores.

## 4. Discussion

ACS, regardless of the reason for its onset, impacts a patient’s physical and psychological life, with repercussions on a subject’s general health and quality of life [9,12,13]. However, as indicated above, not many studies have analyzes the impact of physical limitations on psychological aspects such as quality of life, anxiety, depression, or social functioning, and even less on post-traumatic growth. Therefore, this study aimed to fill this gap. The results showed significant differences among the three groups not only immediately after the ACS episode but also 3 months after it. Significant differences were observed for many variables in physical, mental, and emotional health, which contribute to define the more general concept of quality of life. Thus, on the basis of the results of the Bruce test, differences among the three groups clearly emerged. Patients who managed to achieve a level of more than 10 METS and were therefore classified as functionally better for performing activities requiring physical effort, also scored higher in general health, physical function, vitality, social function, body pain, than the groups of patients who obtained a functional classification level between 7 and 9 METS or less than 6 METS. In essence, the group of patients classified as the worst had a higher degree of limitations, with obvious repercussions on their quality of life in general (higher levels of depression and anxiety, poorer physical and mental health and sexual relationships, higher distress, and lower adjustment to disease). In this sense, future studies analyzing patients’ quality of life after ACS should consider how physically functional patients are when determining the impact of ACS on their psychological health and quality of life. In relation to this, it should be noted that the physical aspects are related to the psychological aspects and vice versa. This is important for the interpretation of the directionality of the results. In this study, we discuss the results considering that physical functional difficulties influence psychological aspects, but we have to be aware that physical limitations can be influenced by psychological aspects.

These results have a deep effect on the understanding of a patient’s situation and, above all, on the perception of the inevitable change in life that the patient will experience. This change has an impact because it leads to a redefinition of the patient’s role in everyday life, in the family and in the workplace [28]. This study shows, in fact, different results for the three considered groups, probably in line with their perception of the change they will experience. These results suggest a strong link between the two components of health, the purely physical and the mental one, which underlines their reciprocal influence. In our case, the results are in line with our starting hypothesis and with the data reported in the literature that show a significant influence of physical function on mental health. In particular, a lower level of physical function would negatively influence all the variables that contribute to general health [22].

However, considering the evolution of the variables after three months, our results reveal differences compared to other studies. In fact, many studies in the literature show a worsening of almost all variables we analyzed; in particular, the various follow-up studies describe a clear worsening in the general state of health and in the quality of life of the patients, above all, in the levels of anxiety and depression [29]. One possible reason for this discrepancy could be the fact that new interventions have become available to cardiologists in the last decades, which are reducing the impact of this trauma by means of less invasive cardiac interventions and the reduction of the days of stay in the hospital.

Another interesting result is that in all three groups of patients included in the study, there was a positive change observed for some variables (improving their physical and mental health, total adjustment, anxiety, and depression and even higher scores in posttraumatic growth). Our interpretation is that this positive change is probably related to the willingness, or need, for a change in lifestyle to overcome the acute phase of the disease as well as to the indications of the medical staff on the management of the trauma. Thus, it appears that a heart attack is very often perceived as a starting point for a lifestyle change [30]. This process of adaptation after a traumatic event is referred to as “benefit finding”, indicating the possibility for a patient to find the positive side of a negative experience. In many diseases (cancer, human immunodeficiency virus, acquired immune deficiency syndrome), this important factor has been found [31]. Another possible explanation could be related to the time of measurement. Many studies report low quality of life after one year after ACS; in our study, quality of life was measured after 3 months. We consider this difference to be important because this period is close to the cardiac event and could influence whether patients are receiving care and social support from relatives. In this sense, an important factor for quality of life after cardiac events is the social support received, as indicated recently [18].

For ACS and many other heart diseases, further studies are needed. In fact, from our results, it appears that, after three months, patients with less limitations improved their scores of many quality-of-life variables in comparison with the groups with higher limitations. Specifically, the improvement in posttraumatic growth is interesting because patients with less limitations significantly increased their scores in appreciation of life change, relationships, and personal strength in comparison to the other groups (life appreciation improved only for group B). Therefore, the lesser the limitations, the higher the posttraumatic growth. This result should open an interesting field of study, related to individual differences in the interpretation of a traumatic event such as ACS. In this sense, patients with greater physical limitations should have a more directive approach to improving their physical function to reduce a trauma’s impact. On the other hand, psychological therapies, such as mindfulness and acceptance and commitment therapy, could reduce the psychological impact of ACS.

Some limitations warrant a cautious interpretation of our results. The most evident limitation of this study is the low number of participants in comparison to other studies and the lack of a control group enabling to determine if the analyzed variables were influenced by the traumatic event or if other factors impacted the patients’ lives. This limitation reduces the generalizability of our results, considering that the data were collected only in one hospital. Moreover, we consider the lack of a stress test after the three months to compare the results obtained in the first measure as a limitation. Another limitation of the study is that the patients were not subjected to the tests within the same period, and this could have influenced their responses. From a statistical point of view, another limitation is the number of contrasts used to compare the samples which, despite showing significant differences, increased the probability of committing a type I error. Future studies must overcome these limitations.

## 5. Conclusions

In conclusion, and bearing in mind the limitations of this study, physical limitations after ACS seem to affect variables such as general health, vitality, total adaptation, emotional role, social adaptation, depression, and anxiety. Of relevance is the fact that the group of subjects classified as worse affected and therefore with a higher degree of limitations, reported lower scores for the variables considered, while the group of patients with a normal level of functionality, without any limitation, showed better scores immediately after ACS diagnosis as well as after 3 months. Moreover, a very interesting result concerns the improvement that emerged in the follow-up for almost all variables considered in all three groups of subjects. This can be attributed to different factors, including the so-called “finding benefit” effect. Further studies are needed to substantiate these results and to investigate which factors influence the improvement in patients’ quality of life perception. Finally, based on our results, we remark that psychological intervention in ACS patients should consider that patients with higher limitations would deserve more psychological attention in relation to their limitations than patients that can return to their daily life without physical restrictions.

## Figures and Tables

**Table 1 ijerph-18-06473-t001:** Mean and standard deviation in the three groups of the following variables: Age (in years), days in the hospital from the admittance to hospital to discharge, body mass index (BMI), systolic and diastolic blood pressure (SBP and DBP), heart rate (HR), cholesterol HDL and LDL, level of triglycerides, glomerular filter and glycate.

	>10 METS (n = 96)		7–9 METS (n = 26)		<6 METS (n = 24)	
	MEAN	SD	MEAN	SD	MEAN	SD
Age	55.34	8.21	61.42	8.64	64.04	11.09
Hospital days	4.69	3.32	4.38	2.10	5.58	3.19
BMI	28.42	3.53	29.41	4.38	28.65	3.10
SBP	127.36	17.77	138.23	25.40	135.83	20.19
DBP	76.89	10.18	75.00	10.87	72.71	12.83
HR	60.65	9.93	59.96	11.61	64.46	12.19
LDL cholesterol	124.75	41.72	114.69	35.01	102.92	31.48
HDL cholesterol	40.20	8.83	39.77	10.54	40.25	11.77
Tryglicerides	154.32	63.89	143.15	60.17	121.54	58.55
Glomerular filter	89.07	16.96	78.67	17.22	95.21	32.34
Glycate	5.94	0.85	6.30	1.37	6.30	1.48

**Table 2 ijerph-18-06473-t002:** Mean and standard deviation in the three groups of the following tests: SF-12, PAIS, HAD and PTGI. Two last columns report statistical differences between groups (first column after ACS and the second after 3 months).

	>10 METS (n = 96) (A)				7–9 METS (n = 24) (B)				<6 METS (n = 24) (C)				DIFFERENCES After ACS	DIFFERENCES After 3 Months
	After ACS		After 3 Months		After ACS		After 3 Months		After ACS		After 3 Months			
	MEAN	(SD)	MEAN	(SD)	MEAN	(SD)	MEAN	(SD)	MEAN	(SD)	MEAN	(SD)		
	**SF-12**													
GENERAL HEALTH	50.52	16.21	52.86	17.35	45.19	17.35	47.12	20.40	36.46	16.45	43.75	15.20	H = 14.209, *p* = 0.001 A–C	H = 6.176, *p* = 0.046 A–C
PHYSICAL FUNCTION	68.75	31.83	74.74	31.73	71.15	26.17	74.04	30.40	41.67	26.24	51.04	35.72	H = 16.618, *p* = 0.000 A–C, B–C	H = 9.592, *p* = 0.008 A–C, B–C
PHYSICAL ROLE	73.44	42.27	76.56	40.36	69.23	47.07	78.85	40.43	45.83	41.49	33.33	45.84	H = 9.817, *p* = 0.007 A–C, B–C	H = 19.341, *p* = 0.000 A–C, B–C
EMOTIONAL ROLE	84.38	32.70	81.25	36.45	73.08	40.57	71.15	42.83	45.83	48.72	39.58	46.58	H = 16.192, *p* = 0.000 A–C, B–C	H = 18.443, *p* = 0.000 A–C, B–C
BODY PAIN	89.58	19.76	90.63	18.58	82.69	20.94	82.69	23.20	71.88	23.67	68.75	26.83	H = 16.873, *p* = 0.000 A–B, A–C	H = 20.255, *p* = 0.000 A–B, A–C, B–C
VITALITY	64.79	22.43	70.00	22.85	66.15	20.99	67.69	21.97	46.67	29.88	48.33	29.44	H = 9.172, *p* = 0.010 A–C, B–C	H = 12.168, *p* = 0.002 A–C, B–C
SOCIAL FUNCTION	83.07	20.36	83.85	19.86	78.85	20.85	77.88	22.72	59.38	26.39	63.54	25.52	H = 16.730, *p* = 0.000 A–C, B–C	H = 13.576, *p* = 0.001 A–C, B–C
MENTAL HEALTH	70.31	21.15	71.88	20.23	66.92	20.35	69.23	20.77	55.42	24.31	55.42	22.06	H = 7.999, *p* = 0.018 A–C	H = 11.261, *p* = 0.004 A–C, B–C
PHYSICAL HEALTH	70.57	22.03	73.70	20.76	67.07	23.09	70.67	20.52	48.96	21.23	48.96	21.23	n.s	n.s.
MENTAL HEALTH	75.64	20.94	76.74	20.92	71.25	22.03	71.49	23.57	51.82	28.15	51.72	26.90	n.s.	n.s.
	**PAIS**													
HEALTH ATTITUDE	0.47	0.32	0.33	0.26	0.44	0.35	0.31	0.28	0.59	0.31	0.41	0.22	n.s.	n.s.
PROFESSIONAL CONTEXT	0.49	0.48	0.50	0.52	0.68	0.59	0.64	0.65	0.81	0.53	0.85	0.63	n.s.	n.s.
DOMESTIC CONTEXT	0.19	0.29	0.17	0.30	0.25	0.32	0.23	0.33	0.29	0.23	0.27	0.27	n.s.	n.s.
SEXUAL RELATIONS	0.49	0.53	0.46	0.55	0.52	0.51	0.53	0.53	0.79	0.41	0.83	0.49	H = 8.082, *p* = 0.018 A–C	H = 10.344, *p* = 0.006 A–C, B–C
FAMILY RELATIONS	0.09	0.19	0.08	0.19	0.15	0.31	0.13	0.33	0.11	0.18	0.11	0.16	n.s.	n.s.
SOCIAL CONTEXT	0.40	0.55	0.37	0.53	0.49	0.48	0.48	0.51	0.49	0.55	0.43	0.55	n.s.	n.s.
DISTRESS	0.40	0.40	0.30	0.41	0.49	0.42	0.40	0.38	0.56	0.34	0.39	0.35	H = 6.448, *p* = 0.040 A–C	H = 5.987, *p* = 0.050 A–C
TOTAL ADJUSTMENT	0.36	0.28	0.31	0.27	0.42	0.32	0.38	0.32	0.49	0.18	0.43	0.21	H = 7.951, *p* = 0.019 A–C	H = 6.997, *p* = 0.030 A–C
	**HAD**													
ANXIETY	3.78	3.18	3.06	3.22	5.27	3.69	4.58	3.70	5.96	2.99	4.83	3.34	H = 10.750, *p* = 0.005 A–C	H = 8.122, *p* = 0.017 A–C
DEPRESSION	2.59	2.58	2.38	2.59	4.04	3.05	3.92	3.61	4.71	3.44	3.88	3.31	H = 11.006, *p* = 0.004 A–C	H = 6.502, *p* = 0.039 A–B, A–C
	**ICP**													
APPRECIATION LIFE CHANGE	2.15	1.20	2.34	1.25	2.25	1.41	2.48	1.45	2.40	1.38	2.67	1.44	n.s.	n.s.
RELATIONS CHANGE	14.52	7.99	15.97	8.61	16.58	9.02	17.96	9.70	16.54	8.22	17.67	8.73	n.s.	n.s.
PERSONAL STRENGH CHANGE	9.97	5.69	11.29	6.41	11.31	6.52	12.08	7.17	11.75	6.08	12.42	6.30	n.s.	n.s.
SPIRITUAL CHANGE	4.40	3.26	4.36	3.25	5.23	3.64	5.46	3.86	5.13	3.57	5.67	3.80	n.s.	n.s.

A–B = significant differences between groups A and B (*p* < 0.05); A–C = significant differences between groups A and C (*p* < 0.01); B–C = significant differences between groups B and C (from (*p* < 0.05 to *p* < 0.001).

**Table 3 ijerph-18-06473-t003:** Statistical differences between the measures at two times after ACS within each group.

	>10 METS (n = 96)		7–9 METS (n = 24)		<6 METS (n = 24)	
	After ACS	After 3 Months	After ACS	After 3 Months	After ACS	After 3 Months
	**SF-12**					
GENERAL HEALTH	Z = −2.49, *p* = 0.013		n.s.		Z = −2.008, *p* = 0.045	
PHYSICAL FUNCTION	Z = −3.83, *p* < 0.001		n.s.		Z = −2.64, *p* = 0.008	
PHYSICAL ROLE	n.s.		n.s.		n.s.	
EMOTIONAL ROLE	n.s.		n.s.		n.s.	
BODY PAIN	n.s.		n.s.		n.s.	
VITALITY	Z = −3.72, *p* < 0.001		n.s.		n.s.	
SOCIAL FUNCTION	n.s.		n.s.		n.s.	
MENTAL HEALTH	Z = −2.39, *p* = 0.017		n.s.		n.s.	
PHYSICAL HEALTH	Z = −3.93, *p* < 0.001		n.s.		n.s.	
MENTAL HEALTH	Z = −203, *p* = 0.042		n.s.		n.s.	
	**PAIS**					
HEALTH ATTITUDE	Z = −6.40, *p* < 0.001		Z = −4.13, *p* = 0.001		Z = −3.77, *p* < 0.001	
PROFESSIONAL CONTEXT	n.s.		n.s.		n.s.	
DOMESTIC CONTEXT	Z = −2.38, *p* = 0.017		n.s.		n.s.	
SEXUAL RELATIONS	Z = −1.92, *p* = 0.05		n.s.		n.s.	
FAMILY RELATIONS	n.s.		n.s.		n.s.	
SOCIAL CONTEXT	n.s.		n.s.		n.s.	
DISTRESS	Z = −4.96, *p* < 0.001		Z = −2.26, *p* = 0.024		Z = −2.87, *p* = 0.004	
TOTAL ADJUSTMENT	Z = −5.42, *p* < 0.001		Z = −2.63, *p* = 0.008		Z = −3.16, *p* = 0.002	
	**HAD**					
ANXIETY	Z = −4.48, *p* < 0.001		Z = −2.27, *p* = 0.023		Z = −2.70, *p* = 0.007	
DEPRESSION	Z = −1.91, *p* = 0.05		n.s.		Z = −2.88, *p* = 0.004	
	**ICP**					
APPRECIATION LIFE CHANGE	Z = −2.48, *p* = 0.013		Z = −2.36, *p* = 0.018		n.s.	
RELATIONS CHANGE	Z = −3.37, *p* = 0.001		n.s.		n.s.	
PERSONAL STRENGH CHANGE	Z = −4.13, *p* = 0.001		n.s.		n.s.	
SPIRITUAL CHANGE	n.s.		n.s.		n.s.	

n.s. means not significant.

## Data Availability

The data presented in this study are available on request from the corresponding author.

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
