# Peer review of "Psychological State after an Acute Coronary Syndrome: Impact of Physical Limitations"

_ijerph, 2021, doi:10.3390/ijerph18126473_

Round 1

Reviewer 1 Report

Although the topic is of clinical interest, the authors did not conduct the research with proper controls.

Specific comments:

  1. The in-text citation style is not in accordance with the journal's guidelines. In the text, reference numbers should be placed in square brackets [ ], and placed before the punctuation; for example [1], [1–3] or [1,3].
  2. Please change "were no met" to "were not met".
  3. How was the sample size determined? There is currently no evidence of power calculation.
  4. "All measures used in this study are validated measures in the clinical context" - at least a citation is necessary here.
  5. Poor physical health can lead to an increased risk of developing mental health problems. Similarly, poor mental health can negatively impact on physical health. Better baseline physical health is also positively correlated with physical activity which has a positive impact on current mental health. Many of these confounding factors were not adequately discussed.
  6. Selection bias and lack of generalizability of findings were potential study limitations but were not discussed by the authors.

Author Response

Reviewer 1

Although the topic is of clinical interest, the authors did not conduct the research with proper controls.

RESPONSE: We are aware of this limitation as we have reflected it in the limitations section. However, the circumstances in which the study was conducted (given the lack of funding) did not allow for a control group.

Specific comments:

  1. The in-text citation style is not in accordance with the journal's guidelines. In the text, reference numbers should be placed in square brackets [ ], and placed before the punctuation; for example [1], [1–3] or [1,3].

RESPONSE: Thank you for your comment. The references have been arranged according to the specifications of the magazine.

  1. Please change "were no met" to "were not met".

RESPONSE: This expression has been changed.

  1. How was the sample size determined? There is currently no evidence of power calculation.

RESPONSE: The sample size was not calculated statistically because we had a time constraint for the study (one year and without funding). Thus, we offered all patients who passed through the hospital to participate in the study and the final study sample was the one who agreed to participate. We are aware that other similar studies have larger samples. Therefore, taking into account your comments, we have included among the limitations that a larger number of subjects would be desirable to test the hypotheses.

  1. "All measures used in this study are validated measures in the clinical context" - at least a citation is necessary here.

RESPONSE: Several citations have been given to justify this phrase

  1. Poor physical health can lead to an increased risk of developing mental health problems. Similarly, poor mental health can negatively impact on physical health. Better baseline physical health is also positively correlated with physical activity which has a positive impact on current mental health. Many of these confounding factors were not adequately discussed.

RESPONSE: Thank you for your comment. We have included this idea in the discussion. In addition, in order to be more cautious in our conclusions we have tried to qualify that our results may be affected by these factors.

  1. Selection bias and lack of generalizability of findings were potential study limitations but were not discussed by the authors.

RESPONSE: As in the previous comment, we have discussed these factors in more detail. In addition, in order to be more cautious in our conclusions we have tried to qualify that our results may be affected by these factors.

Reviewer 2 Report

This study has the potential to be of significant interest as a contribution to our understanding of the psychological impact which follows acute coronary occlusions. Coronary occlusions have considerable psychosocial consequences in many patients because of the obvious risks of death and future morbidity. The authors have stratified their study subjects by a measure of physical impairment following ACS and have then measured a large number of psychosocial parameters around the time of presentation to a hospital and again after three months. Their findings confirmed their hypothesis that the severity of physical impairment at admission predicted important elements of psychosocial health, both immediately and after a further three months. They also observed some positive changes in psychological health characterised by more positive and constructive attitudes to life.

Unfortunately the authors have written with a markedly verbose style of language which has made reading their paper difficult. They need to be much more select in the information they provide and be much more concise in their style. I suggest that they begin by summarising the essential reasons for conducting this study, the rationale for the design they used and the core questions they wanted to answer through this project. Their hypotheses are reasonable but cover several areas of psychosocial function. Namely, psychological symptoms (eg anxiety, depression), psychological function (eg quality of life), adjustment to ill health (eg attitudes to illness) and positive life changes (eg greater life appreciation). I suggest that these need to be distinctly identified as themes in the paper and the results presented in three or more groups. The measures used should be presented succinctly and their purpose made clear. I wonder if the authors have been too ambitious in adding a measure of "personal growth" to the other aspects of illness behaviour, or may even prefer to make this the prime focus of their study.

The data analysis is logical but does not offer a correction for the large number of measures which are obtained from this one study group. A Bonferroni correction or similar statistical analysis would be appropriate and I suggest that a multivariate analysis of some kind could be very helpful. The presentation of the results is very difficult to follow and I suggest that several tables with the inclusion of basic statistical values (eg measures of significance and effect size). Providing all of this data in the text is overwhelming, given the number of variables. Summaries of each data set in the text would also be very useful. 

It is not clear why inter-group differences were analysed for the initial measures but not those at 3 month follow-up. These findings would be of considerable interest.

I suggest that clinical experience and previous research would have suggested acute coronary syndrome events would very likely have adverse psychological effects upon patients. Variations in such effects are clearly of interest. Correlations between physical impairment post-ACS and psychosocial health are also clearly of interest and a positive interaction most intuitively expected. However I found the observation of personal growth and constructive changes in psychosocial functioning to be of particular interest. These effects were not unexpected but far less intuitively predictable. A more precise summary and comment upon these findings would be of value, particularly if such changes could be linked to aspects of post-ACS physical disability (which is the focus of this study).

Author Response

This study has the potential to be of significant interest as a contribution to our understanding of the psychological impact which follows acute coronary occlusions. Coronary occlusions have considerable psychosocial consequences in many patients because of the obvious risks of death and future morbidity. The authors have stratified their study subjects by a measure of physical impairment following ACS and have then measured a large number of psychosocial parameters around the time of presentation to a hospital and again after three months. Their findings confirmed their hypothesis that the severity of physical impairment at admission predicted important elements of psychosocial health, both immediately and after a further three months. They also observed some positive changes in psychological health characterised by more positive and constructive attitudes to life.

Unfortunately the authors have written with a markedly verbose style of language which has made reading their paper difficult. They need to be much more select in the information they provide and be much more concise in their style. I suggest that they begin by summarising the essential reasons for conducting this study, the rationale for the design they used and the core questions they wanted to answer through this project.

RESPONSE: Thank you for your comment. We have revised the whole manuscript trying to be more concise and direct in the way of expression in order to better understand the ideas expressed. We hope that the new version of the manuscript will be better written and more understandable.

Their hypotheses are reasonable but cover several areas of psychosocial function. Namely, psychological symptoms (eg anxiety, depression), psychological function (eg quality of life), adjustment to ill health (eg attitudes to illness) and positive life changes (eg greater life appreciation). I suggest that these need to be distinctly identified as themes in the paper and the results presented in three or more groups. The measures used should be presented succinctly and their purpose made clear. I wonder if the authors have been too ambitious in adding a measure of "personal growth" to the other aspects of illness behaviour, or may even prefer to make this the prime focus of their study.

RESPONSE: Thank you again for your comment. We have revised the introduction, hypothesis, results and discussion based on your suggestion to try to better adjust the message and make it more organised. Hopefully it will appear more organised and clearer.

The data analysis is logical but does not offer a correction for the large number of measures which are obtained from this one study group. A Bonferroni correction or similar statistical analysis would be appropriate and I suggest that a multivariate analysis of some kind could be very helpful.

RESPONSE: We agree with your observation. We have tried to apply statistical corrections, but once applied the results, unfortunately, diminish in strength. Moreover, we have consulted with statistical experts and they do not recommend multivariate analysis due to the non-normality of the variables, so we have to do non-parametric statistics, which does not allow multivariate analysis, and forces us to make multiple statistical comparisons. Bearing in mind that we do not want to give up on the results presented, since we consider them interesting despite the low statistical power, the only solution we have found is to indicate this as a limitation and to be more cautious when discussing the results. Although we know that this is not the best solution, we do believe that in this way the potential reader will be able to properly interpret the study and its results. 

The presentation of the results is very difficult to follow and I suggest that several tables with the inclusion of basic statistical values (eg measures of significance and effect size). Providing all of this data in the text is overwhelming, given the number of variables. Summaries of each data set in the text would also be very useful. 

RESPONSE: As indicated above, we have rewritten the results to focus more on the meaning rather than the test used.

It is not clear why inter-group differences were analysed for the initial measures but not those at 3 month follow-up. These findings would be of considerable interest.

RESPONSE: We do not quite understand this comment. All possible tests have been carried out and reported.

I suggest that clinical experience and previous research would have suggested acute coronary syndrome events would very likely have adverse psychological effects upon patients. Variations in such effects are clearly of interest. Correlations between physical impairment post-ACS and psychosocial health are also clearly of interest and a positive interaction most intuitively expected.

RESPONSE: Thank you for your comment. The results have been discussed further given the interest they present. Hopefully the current version of the manuscript will be fine.

However I found the observation of personal growth and constructive changes in psychosocial functioning to be of particular interest. These effects were not unexpected but far less intuitively predictable. A more precise summary and comment upon these findings would be of value, particularly if such changes could be linked to aspects of post-ACS physical disability (which is the focus of this study).

RESPONSE: We have discussed further the issue of personal growth as you suggest. For us it is also an important issue and we feel that it is poorly addressed in medicine when suffering from serious illnesses such as ACS.

Reviewer 3 Report

This is a well constructed study. The manuscript, however, needs work. I am uploading my comments within the pdf form because I thought that would be easier. I did not do as much commenting within the discussion because it should be revised (including separating it into formative paragraphs). Be sure to double check your citations and references (note line 414 reference).

Author Response

Reviewer 3

This is a well constructed study. The manuscript, however, needs work. I am uploading my comments within the pdf form because I thought that would be easier. I did not do as much commenting within the discussion because it should be revised (including separating it into formative paragraphs). Be sure to double check your citations and references (note line 414 reference).

RESPONSE: Your comments and corrections to the pdf are very much appreciated. We have made all the proposed changes and reorganised the information as you have indicated. The references have also been arranged according to the journal's standards.

Round 2

Reviewer 1 Report

Thank you for the revisions.

Reviewer 2 Report

I note that the authors have addressed the concerns I raised in my previous review very carefully and constructively. The paper is now easy to read and therefore presents the author's rationale for the research coherently and explains the results obtained constructively. They have more clearly addressed the limitations of their methodology and the inherent limitations of this form of research. In addition they have placed greater emphasis upon the more significant elements of their findings and therefore raised the value of their research. I have no further recommendations regarding this paper.